# Hepatitis B and HIV coinfection in Northern Uganda: Is a decline in HBV prevalence on the horizon?

Annacarla Chiesa[1], Emmanuel Ochola[2,3]*, Letizia Oreni[1], Paolo Vassalini[1,4], Giuliano Rizzardini[5], Massimo Galli[1]

1 Infectious Disease Unit, L. Sacco Department of Biomedical and Clinical Sciences, University of Milan, Milan, Italy, 2 Department of HIV, Research and Documentation, St. Mary's Hospital Lacor, Gulu, Uganda, 3 Department of Public Health, Gulu University Faculty of Medicine, Gulu, Uganda, 4 Department of Public Health and Infectious Diseases, Sapienza University of Rome, Rome, Italy, 5 First Infectious Disease Division, Fatebenefratelli Sacco Hospital, Milan, Italy

* ayiga33@hotmail.com

## Abstract

### Background

The available data concerning hepatitis B virus (HBV) infection in Uganda are limited, particularly in the case of people living with HIV/AIDS (PLWH). HBV is not routinely tested when starting antiretroviral therapy (ART). We aimed to determine the prevalence, the correlates of the risk of HBV infection, and the association with outcomes of ART among PLWH attending a busy HIV clinic in a referral hospital in Northern Uganda.

### Patients and methods

From April to June 2016, a random sample of 1000 PLWH attending the outpatients' clinic of St. Mary's Hospital, Gulu, Uganda were systematically selected to undergo a rapid hepatitis B surface antigen (HBsAg) test after administering a questionnaire in this cross-sectional study. HIV care parameters were obtained from client files. Multivariate logistic regression and general linear model were used for the analysis.

### Results

950 of the 985 evaluable patients (77% females; mean age 42.8 years) were receiving ART. The overall prevalence of HBsAg was 7.9% (95% confidence interval [CI] 6.2–9.6%), and was significantly lower among the females (6.8% *vs* 11.7%; p = 0.020). The factors independently associated with higher HBV infection were having lived in an internally displaced persons' camp (adjusted odds ratio [aOR] 1.76, 95% CI 1.03–2.98; p = 0.036) and having shared housing with HBV-infected people during childhood (aOR 3.30, 95% CI 1.49–7.32; p = 0.003). CD4+ T cell counts were significantly lower in HBV patients (p = 0.025), and co-infection was associated with a poorer CD4+ T cell response to ART (AOR 0.88; 95% CI 0.79–0.98; p = 0.030).

**Data Availability Statement:** All relevant data are within the  Supporting information files.

**Funding:** AC was supported funding support from Fondazione Piero e Lucille Corti Onlus, which

supported her stay in Uganda, the cost of the test kits and the facilitation of data collectors. https://fondazionecorti.it/ The funders had no role in the study design, data collection and analysis, decision to publish, or preparation of the manuscript.

**Competing interests:** The authors have declared that no competing interests exist.

## Conclusions

The observed prevalence of HBV among the PLWH may be underestimated or a signal of HBV decline in the region. The factors favouring horizontal HBV transmission identified suggest extending HBV screening and vaccine prophylaxis among PLWH.

## Introduction

Approximately 7.4 percent of persons living with HIV (PLWH) worldwide are infected with hepatitis B virus (HBV) with considerable geographical variation depending on overall HBV prevalence in the country [1, 2]. Prevalence of HIV-HBV coinfection is 4–6% in countries with low HBV endemicity and 10–25% in those with intermediate and high endemicity [3]. In areas with low HBV endemicity, both viruses are more often acquired in adulthood through sexual and percutaneous contacts, whereas in hyperendemic horizontal and mother-to-child transmission of HBV predominate [3, 4]. In Africa, the World Health Organisation (WHO) estimates a low/intermediate prevalence of HBV in northern Africa (2–4%), high/intermediate prevalence in Sudan, South Sudan, and central, eastern and southern Africa (5–7%), and high prevalence above eight percent in Chad, Cameroon, and western Africa [1, 2]. However, there is paucity of HIV-HBV co-infection data, although reports show that up to 36% of some cohorts of people living with HIV/AIDS (PLWH) have chronic hepatitis B infection [5].

Bwogi *et al.* [6] reported an 8.3% prevalence of co-infection in Uganda in 2005, while the estimated prevalence of general HIV was 7.4% [7] in 2015 and 6.2% in 2017 [8]. HBV prevalence was estimated at 10.3% [6] in 2005 and 4.1% in 2017 [8]. If these data are accurate, this would signal a large reduction in HBV prevalence in Uganda. Uganda however has wide regional variations in the prevalence of both HIV and HBV infections [6–8]. In Northern Uganda, the reported prevalence of HIV is 3.1–7.2% and that of HBV is 6–12% in the Acholi subregion of Northern Uganda [8] though earlier figures showed 18.5–23,9%. The same study showed a lower HBV prevalence in Southern Uganda of 4.7–8% [6, 8]. Furthermore, as Northern Uganda was the theatre of a civil war from 1989 to 2006 [9], civilians were compelled to seek refuge in camps where hygiene and living conditions were often poor [10]. In the Gulu region of where this study was carried out, an earlier population based survey in 2010 determined the prevalence of HBV infection to be 17.6% (95%CI 14.9–20.3) [11] with no documentation of HIV status. There is insufficient data to show that regional differences in HIV and HBV prevalence correlate with the prevalence of HBV-HIV coinfection.

HBV is a major cause of morbidity and mortality in PLWH [5, 12–14]. PLWH are at higher risk of occult HBV infection than their HIV-negative counterparts and have six times higher risk of developing chronic hepatitis after acute HBV infection [4, 5, 13, 15]. Moreover, co-infected persons more rapidly develop liver fibrosis, cirrhosis, end-stage liver disease and hepatocellular carcinoma (HCC) [3–5, 13, 15]. Coinfected patients have lower CD4+ T cell counts which respond more slowly to antiretroviral therapy (ART) [16–18], and are more likely to develop immune reconstitution inflammatory syndrome (IRIS) and drug-induced liver damage [13, 19] However, CD4+ T cell count rescue in response to ART is better in HIV/HBV patients who develop hepatitis B surface antibodies (HBsAb) than in those who fail to clear HBV [20].

Given the similarities of the enzymatic pocket of HIV reverse transcriptase and HBV DNA polymerase, some HIV-reverse transcriptase inhibitors can target both viruses [2, 15, 21]. However, underdiagnosis of HBV infection can lead to a failure to start PLWH on AART

regimens that are active against HBV and inappropriate ART discontinuation may lead to a flare of HBV infection [13]. The aim of this study was thus to determinate the prevalence of HBV-HIV co-infection among PLWH, the risk factors associated with co-infection, and the association between co-infection and the choice of ART and treatment outcomes among HIV patients in a busy HIV clinic in a referral hospital in Northern Uganda.

## Materials and methods

This cross-sectional study involved 1000 respondents aged 13 years or older, selected by systematic random sampling of patients attending the HIV clinic of Saint Mary's Hospital Lacor, the largest hospital in Northern Uganda, from April to June 2016. The AIDS outpatient clinic had cumulatively cared for 14,962 patients by 2015, of whom 6,217 were actively receiving ART [22]. Inclusion age was chosen because those below 13 years had received HBV vaccination that was introduced into Uganda's vaccination programme in 2002 [23]. Patients requiring emergency care at the time of recruitment were excluded from the study.

Regardless of treatment status, participation in the study was offered to one in every 2–4 patients (depending on the number of visits expected each day) in their order of arrival at the outpatient clinic with the aim of randomly and consecutively recruiting 25 patients a day in order to reach a total of 1000 cases after a pre-fixed series of 40 days, starting on 12th April 2016. Sample size was calculated using the Leslie-Kish formula, a confidence level of 95%, a precision of 1.9%, and the estimated prevalence of HBV infection in Uganda (10.3%) [6]. Written informed consent was obtained from all participants. Respondents aged 13–17 years provided assent in addition to consent by their legal guardians.

A questionnaire was administered by nurses trained to collect participants' age, sex, ethnic group, religion, residence, childhood (type of delivery and HBV status of mother, type of habitation, the number of siblings and cohabiting people, and the presence of HBV-infected persons). Other variables included displacement in an internal refugee camp during war, level of education, type of employment, history of military service and sexual history, circumcision, scarification, the use of injected drugs, blood transfusions, marital status and type of marriage (monogamous/polygamous), number of children, and HIV treatment adherence. Respondents' clinical data (date of HIV diagnosis, ART and AIDS status, CD4+ T cell counts at the time of HIV diagnosis, start of ART, and after 6, 12 and 18 months of therapy, WHO clinical stage) were obtained from the patient records.

Upon enrolment, all participants were tested for HBsAg (One-Step HBsAg Test, InTec Products Inc., Xiamen, China; sensitivity 98.89%, specificity 98.87%), aspartate aminotransferase (AST) and alanine aminotransferase (ALT) (LiquiUV System reagent for HumaStar600, Human Gesellschaft für Biochemica und Diagnostica mbH, Wiesbaden, Germany) in accordance with the standard procedures of the manufacturer adapted at St. Mary's Hospital Lacor.

We also measured CD4+ T cell counts if not tested during the previous six months, and assessed HIV viral load if not tested during the previous twelve months.

The study received ethical approval from the Lacor Hospital Institutional Research and Ethics Committee and the Uganda National Council for Science and Technology (number HS 2034) in accordance with Ugandan regulations. Patient information was kept confidential and was accessible only by the study investigators.

### Statistical analysis

Data were entered in an Excel spreadsheet, and analysed using SAS 9.4 and XLSTAT software. Wald's test was used to calculate the population prevalence of HBV, and bivariate analysis was used to identify risk factors for HBV infection using the χ2 test or Fisher's exact test for

categorical variables, and the Mann-Whitney test for the quantitative variables. The nominal level of significance was set at α = 0.05. Factors associated with HBsAg positivity were analysed using univariate and multivariate logistic regression, and a general linear model was used to assess CD4+ T cell response to ART.

## Results

The study recruited 1000 PLWH over 44 days at the outpatient clinic (a mean of 22 subjects per day); 15 subjects were excluded because r HBsAg was not tested after completing the survey, leaving a total of 985 subjects. 35 subjects not receiving ART would not affect the results of the statistical analysis, these were also excluded, and the final analysis was based on the 950 receiving ART.

The study population was majorly females (736, 77%), Catholic (74.3%), and Acholi ethnic group (92%) as shown in Table 1. Median age was 42 years (IQR: 35.0–49.9). 46% declared illiteracy, and only 17% had had more than a primary school education. Only 17.3% reported having formal or informal employment, the most frequent being "*teacher*" and "*business*", whereas 81.2% worked as peasant farmers or were unemployed. A history of military service was infrequent (5.5%), while 49.9% of the study participants had stayed in an internally displaced persons (IDP) camp during the insurgency of the Lord's Resistance Army (LRA).

Most participants were currently or previously married; only 9.6% had "*never married*". Half of all marriages were monogamous, and half were polygamous. 92% of the participants reported having children (five or more in 37.79% of cases). Multiple sexual partners and traditional scarification for therapeutic purposes were reported by 11.3% and 33.4% of the respondents respectively. Only 2.1% of the participants were circumcised, and 11.9% reported receiving a blood transfusion (Table 1).

Almost all respondents said that they were born vaginally, had at least one sibling, and grew up sharing their hut/house with at least two other people. When asked whether anyone was infected with HBV in the compound in which they spent their childhood, 5.8% answered affirmatively and 73.4% answered negatively. Only 0.7% answered that their mother was positive for HBV (Table 1).

Median time from ART initiation to study enrolment was 71.5 months (IQR: 40.6–108.8), and almost all recruited participants were receiving ART (950/985); 67.2% had an undetectable HIV viral load, below 20 copies/ml. Median CD4+ T cell count increased from 240 cells/μL (IQR: 144–325 cells/μL) at the time of starting ART to 521 (IQR:380–695 cells/μL) at most recent available test. Median AST and ALT levels were 32 U/L (IQR:26–39) and 24 U/L (IQR:19–32) respectively (Table 2). Most respondents were on tenofovir (TDF)-containing regimens. 87.4% of the respondents denied missing any doses of therapy.

Seventy-five of the ART-treated patients were HBsAg positive, a prevalence of 7.9% (95% CI:6.2–9.6%). Table 3 shows the predictors of HBsAg positivity. Prevalence of HBsAg was significantly higher among males (11.7% *vs* 6.8%; P = 0.020), but was not significantly influenced by age, ethnicity or religion. It was also significantly higher among the patients who reported growing up in the same compound as an HBV-positive person (P = 0.002), and among those who had been interned in an IDP camp (P = 0.039). Additional gender-stratified analysis did not significantly change the results.

All variables with a P-value of < 0.2 at univariate analysis were included in a multivariate analysis (Table 3), under which sex lost its statistical significance. Having lived in an IDP camp (adjusted odds ratio (AOR) 1.76, 95% CI 1.03–2.98; P = 0.036) or in close contact with a HBV-infected person during childhood (AOR 3.30, 95% CI 1.49–7.32; P = 0.003) however proved to

**Table 1. Socio-economical and behavioural characteristics of the 950 respondents.**

| | TOTAL | | HBsAg positive | | HBsAg negative | | P value* |
|---|---|---|---|---|---|---|---|
| | N = 950 | | n = 75 | | n = 875 | | |
| **Age in years, median (IQR)** | | | | | | | 0.368 |
| | 42.8 | (35.0–49.9) | 42.4 | (33.7–48.2) | 42.8 | (35.4–50.0) | |
| **Gender, N (%)** | | | | | | | ***0.020*** |
| Male | 214 | (22.53) | 25 | (33.33) | 189 | (21.60) | |
| Female | 736 | (77.47) | 50 | (66.67) | 686 | (78.40) | |
| **Religion, N (%)** | | | | | | | 0.139 |
| Catholic | 706 | (74.32) | 56 | (74.67) | 650 | (74.29) | |
| Anglican/Protestant | 142 | (14.95) | 14 | (18.67) | 128 | (14.63) | |
| Pentecostal | 45 | (4.74) | 5 | (6.67) | 40 | (4.57) | |
| Muslim | 12 | (1.26) | 0 | (0.00) | 12 | (1.37) | |
| Other | 45 | (4.74) | 0 | (0.00) | 45 | (5.14) | |
| **Tribe, N (%)** | | | | | | | 0.134 |
| Acholi | 875 | (92.11) | 66 | (88.00) | 809 | (92.46) | |
| Lango | 35 | (3.68) | 6 | (8.00) | 29 | (3.31) | |
| Other | 40 | (4.21) | 3 | (4.00) | 37 | (4.23) | |
| **Level of education, N (%)** | | | | | | | 0.475 |
| No formal education | 438 | (46.11) | 28 | (37.33) | 410 | (46.86) | |
| Completed primary school | 334 | (35.16) | 30 | (40.00) | 304 | (34.74) | |
| Completed senior school | 123 | (12.95) | 13 | (17.33) | 110 | (12.57) | |
| Qualification afther senior six | 38 | (4.00) | 3 | (4.00) | 35 | (4.00) | |
| ND[+] | 17 | (1.79) | 1 | (1.33) | 16 | (1.83) | |
| **Employment, N (%)** | | | | | | | 0.375 |
| Unemployed | 267 | (28.11) | 23 | (30.67) | 244 | (27.89) | |
| Peasant farmer | 410 | (43.16) | 26 | (34.67) | 384 | (43.89) | |
| Informal employment | 152 | (16.00) | 17 | (22.67) | 135 | (15.43) | |
| Formal employment | 108 | (11.37) | 8 | (10.67) | 100 | (11.43) | |
| ND | 13 | (1.37) | 1 | (1.33) | 12 | (1.37) | |
| **Soldier, N (%)** | | | | | | | |
| Yes | 52 | (5.47) | 3 | (4.00) | 49 | (5.60) | 0.559 |
| No | 898 | (94.53) | 72 | (96.00) | 826 | (94.40) | |
| **Type of marriage, N (%)** | | | | | | | 0.144 |
| Polygamous | 429 | (45.16) | 27 | (36.00) | 402 | (45.94) | |
| Monogamous | 429 | (45.16) | 42 | (56.00) | 387 | (44.23) | |
| Never married / ND | 92 | (9.68) | 6 | (8.00) | 86 | (9.83) | |
| **Number of lifetime sex partners, N (%)** | | | | | | | 0.331 |
| ≤ 5 | 843 | (88.74) | 64 | (85.33) | 779 | (89.03) | |
| > 5 | 107 | (11.26) | 11 | (14.67) | 96 | (10.97) | |
| **Number of children, N (%)** | | | | | | | 0.893 |
| 0 | 77 | (8.11) | 7 | (9.33) | 70 | (8.00) | |
| 1–4 | 514 | (54.11) | 41 | (54.67) | 473 | (54.06) | |
| ≥ 5 | 359 | (37.79) | 27 | (36.00) | 332 | (37.94) | |
| **Scarification, N (%)** | | | | | | | 0.804 |
| Yes | 317 | (33.37) | 26 | (34.67) | 291 | (33.26) | |
| No | 633 | (66.63) | 49 | (65.33) | 584 | (66.74) | |
| **Circumcision, N (%)** | | | | | | | 0.066 |
| Yes | 20 | (2.11) | 4 | (5.33) | 16 | (1.83) | |

*(Continued)*

**Table 1.** (Continued)

| | TOTAL | | HBsAg positive | | HBsAg negative | | P value* |
|---|---|---|---|---|---|---|---|
| | **N = 950** | | **n = 75** | | **n = 875** | | |
| No | 930 | (97.89) | 71 | (94.67) | 859 | (98.17) | |
| **Blood transfusion, N (%)** | | | | | | | 0.475 |
| Yes | 113 | (11.89) | 7 | (9.33) | 106 | (12.11) | |
| No | 837 | (88.11) | 68 | (90.67) | 769 | (87.89) | |
| **Lived in a refugee camp, N (%)** | | | | | | | *0.039* |
| Yes | 474 | (49.89) | 46 | (61.33) | 428 | (48.91) | |
| No | 476 | (50.11) | 29 | (38.67) | 447 | (51.09) | |
| **People in the same house/hut during childhood, N (%)** | | | | | | | 0.477 |
| 0–1 | 27 | (2.84) | 4 | (5.33) | 23 | (2.63) | |
| 2–5 | 405 | (42.63) | 32 | (42.67) | 373 | (42.63) | |
| 6–9 | 354 | (37.26) | 25 | (33.33) | 329 | (37.60) | |
| > 9 | 164 | (17.26) | 14 | (18.67) | 150 | (17.14) | |
| **Type of delivery, N (%)** | | | | | | | 0.059 |
| natural childbirth | 875 | (92.11) | 64 | (85.33) | 811 | (92.69) | |
| caesarean section | 21 | (2.21) | 3 | (4.00) | 18 | (2.06) | |
| ND | 54 | (5.68) | 8 | (10.67) | 46 | (5.26) | |
| **HBV-infected person in the house/compound where the respondent spent his/her childhood, N (%)** | | | | | | | *0.002* |
| Yes | 55 | (5.79) | 12 | (16.00) | 43 | (4.91) | |
| No | 698 | (73.47) | 49 | (65.33) | 649 | (74.17) | |
| ND | 197 | (20.74) | 14 | (18.67) | 183 | (20.91) | |
| **Maternal HBV positivity status, N (%)** | | | | | | | 0.610 |
| Yes | 7 | (0.74) | 1 | (1.33) | 6 | (0.69) | |
| No | 635 | (66.84) | 50 | (66.67) | 585 | (66.86) | |
| ND | 308 | (32.42) | 24 | (32.00) | 284 | (32.46) | |

[+]ND: no answer/don't known / not defined.

* p-values are for $\chi 2$ or Fisher's exact test and Mann-Whitney test.

be independent risk factors for HBsAg positivity. A Lower CD4+ T cell count was associated with HBsAg positivity (AOR 0.88, 95% CI: 0.79–0.98; P = 0.030). Rise in CD4+ T cells count in HIV-HBV coinfected patients was significantly less, compared to their HBV negative counterparts(Fig 1 and Table 3), and was independently associated with a higher pre-therapy CD4+ T (β = -0.48, 95% CI -0.54 to -0.42; P<0.0001), male sex (β = -0.15, 95% CI -0.21 to -0.09; P<0,0001) and HBsAg positivity (β = -0.06, 95% CI -0.12 to -0.008; P = 0.027), whereas the response was better the longer the therapy lasted (β = 0.08, 95% CI 0.02–0.14; P = 0.004) (Table 4).

Although their HBV status was unknown before this study was carried out, a higher percentage of HBsAg-positive than HBsAg-negative patients were receiving TDF (P = 0.032), and only four HBsAg-positive patients were being treated with zidovudine+lamivudine (AZT/ 3TC). TDF was included in the first-line ART regimen of 66 of the 68 HBsAg-positive patients currently being treated with it (Table 2).

There was no correlation between HBV coinfection and the other clinical, viral or biochemical variables analysed.

**Table 2. Clinical, virologic, immunological and therapeutic characteristics of the 950 respondents undergoing antiretroviral therapy (ART) by HBV status.**

| | TOTAL | | HBsAg+ | | HBsAg- | | P value* |
|---|---|---|---|---|---|---|---|
| | N = 950 | | n = 75 | | n = 875 | | |
| **WHO stage at ART initiation, N (%)** | | | | | | | 0.833 |
| 1 | 254 | (26.74) | 19 | (25.33) | 235 | (26.86) | |
| 2 | 332 | (34.95) | 25 | (33.33) | 307 | (35.09) | |
| 3 | 252 | (26.53) | 21 | (28.00) | 231 | (26.40) | |
| 4 | 39 | (4.11) | 2 | (2.67) | 37 | (4.23) | |
| ND | 73 | (7.68) | 8 | (10.67) | 65 | (7.43) | |
| **Current AST (U/L)** | | | | | | | 0.057 |
| Median (IQR) | 32 | (26–39) | 34 | (28–43) | 32 | (26–39) | |
| **Current ALT (U/L)** | | | | | | | 0.140 |
| Median (IQR) | 24 | (19–32) | 26 | (20–36) | 24 | (19–32) | |
| **Current HIV viral load undetectable, N (%)** | | | | | | | 0.726 |
| Yes | 639 | (67.26) | 52 | (69.33) | 587 | (67.09) | |
| No | 73 | (7.68) | 4 | (5.33) | 69 | (7.89) | |
| ND | 238 | (25.05) | 19 | (25.33) | 219 | (25.03) | |
| **CD4+ T cell count at ART initiation** | | | | | | | 0.295 |
| Median (IQR) | 240 | (144–325) | 249 | (175–336) | 239 | (142–324) | |
| **CD4+ T cell count at 6 months of ART** | | | | | | | |
| Median (IQR) | 363 | (243–519) | 331 | (240–429) | 368 | (243–520) | 0.422 |
| **CD4+ T cell count at 12 months of ART** | | | | | | | |
| Median (IQR) | 391 | (277–534) | 370 | (295–518) | 395 | (274–538) | 0.742 |
| **CD4+ T cell count at 24 months of ART** | | | | | | | |
| Median (IQR) | 410 | (298–577) | 355 | (288–478) | 421 | (300–580) | 0.143 |
| **Current CD4+ T cell count** | | | | | | | *0.025* |
| Median (IQR) | 521 | (380–695) | 485 | (323–593) | 524 | (385–705) | |
| **Change in CD4+ T cell count (current-initial)** | | | | | | | *0.012* |
| Median (IQR) | 274 | (135–441) | 236 | (87–341) | 280 | (144–448) | |
| **Time of ART initiation, N (%)** | | | | | | | 0.388 |
| 2002–2013 | 782 | (82.32) | 59 | (78.67) | 723 | (82.63) | |
| 2014–2016 | 168 | (17.68) | 16 | (21.33) | 152 | (17.37) | |
| **Current ART regimen, N (%)** | | | | | | | 0.307 |
| 3TC+TDF+PI/NNRTI[+] | 825 | (86.84) | 68 | (90.67) | 757 | (86.51) | |
| Other[++] | 125 | (13.16) | 7 | (9.33) | 118 | (13.49) | |
| **Months of antiretroviral therapy** | | | | | | | 0.129 |
| Median (IQR) | 71.5 | (40.6–108.8) | 61.8 | (36.4–99.3) | 72.1 | (41.3–109.7) | |

ND: no answer/don't known / not defined. IQR: interquartile range. AST/ALT: aspartate aminotransferase/alanine aminotransferase.

NNRTI: Non nucleoside Reverse Transcriptase inhibitor.

[+]3TC: lamivudine, TDF: Tenofovir, PIù: Protease inhibitor.

* p-values are for χ2 or Fisher's exact test and Mann-Whitney test.

[++] The other regimens have lamivudine, which without TDF was not considered as optimally active against HBV.

## Discussion

This study found that HBV infection prevalence among HIV infected patients in a busy hospital clinic in Northern Uganda was 7.9%, and identified three independent factors associated with co-infection: living in an IDP camp between 1989 and 2006 [9, 24], cohabitation with HBV-infected persons during childhood, and a poorer CD4+ T cell count response to ART

**Table 3. Predictors of HBsAg positivity in the population of 950 PWH receiving ART (logistic univariate and multivariate analysis).**

| Univariate | | | | | Multivariate | | | |
|---|---|---|---|---|---|---|---|---|
| | OR | OR lower limit (95%) | OR upper limit (95%) | p-value | OR | OR lower limit (95%) | OR upper limit (95%) | p-value |
| **Gender** | | | | | | | | |
| Female | 1 | | | | 1 | | | |
| Male | *1,815* | 1,094 | 3,011 | *0,021* | 1,574 | 0,867 | 2,857 | 0,136 |
| **Religion** | | | | | | | | |
| Catholic | 1 | | | | 1 | | | |
| Anglican/Protestant | 1,299 | 0,707 | 2,389 | 0,400 | 1,096 | 0,541 | 2,219 | 0,800 |
| Pentecostal | 1,564 | 0,610 | 4,005 | 0,352 | 2,121 | 0,787 | 5,719 | 0,137 |
| Muslim | 0,461 | 0,024 | 8,840 | 0,607 | 0,523 | 0,030 | 9,272 | 0,659 |
| Other | 0,127 | 0,007 | 2,146 | 0,152 | 0,122 | 0,008 | 1,954 | 0,137 |
| **Tribe** | | | | | | | | |
| Acholi | 1 | | | | 1 | | | |
| Lango | *2,536* | 1,017 | 6,327 | *0,046* | 2,317 | 0,829 | 6,480 | 0,109 |
| Other | 0,994 | 0,298 | 3,310 | 0,992 | 1,170 | 0,303 | 4,515 | 0,820 |
| **Type of marriage** | | | | | | | | |
| Monogamous | 1 | | | | 1 | | | |
| Polygamous | 0,619 | 0,374 | 1,024 | 0,062 | 0,852 | 0,488 | 1,487 | 0,573 |
| never married/ND | 0,643 | 0,265 | 1,560 | 0,329 | 1,168 | 0,475 | 2,872 | 0,734 |
| **Circumcision** | | | | | | | | |
| No | 1 | | | | 1 | | | |
| Yes | 3,025 | 0,985 | 9,289 | 0,053 | 1,318 | 0,292 | 5,946 | 0,719 |
| **Refugee camp** | | | | | | | | |
| No | 1 | | | | 1 | | | |
| yes | *1,657* | 1,022 | 2,686 | *0,041* | *1,760* | 1,039 | 2,982 | *0,036* |
| **Type of delivery** | | | | | | | | |
| natural childbirth | 1 | | | | 1 | | | |
| caesarean section | 2,112 | 0,606 | 7,360 | 0,241 | 2,076 | 0,532 | 8,108 | 0,293 |
| ND | 2,204 | 0,997 | 4,869 | 0,051 | 1,552 | 0,602 | 4,004 | 0,363 |
| **HBV in the compound** | | | | | | | | |
| no | 1 | | | | 1 | | | |
| yes | *3,696* | 1,831 | 7,463 | *<0,001* | *3,309* | 1,495 | 7,324 | *0,003* |
| ND | 1,013 | 0,547 | 1,876 | 0,967 | 0,786 | 0,399 | 1,548 | 0,486 |
| **Current ALT (x 1 U/L)** | | | | | | | | |
| | 1,006 | 0,997 | 1,015 | 0,221 | 1,003 | 0,993 | 1,012 | 0,609 |
| **ΔCD4+ T cell (x 100 cells/mm$^3$)** | | | | | | | | |
| | *0,910* | 0,840 | 0,986 | *0,022* | *0,884* | 0,791 | 0,988 | *0,030* |
| **Months of antiretroviral therapy (x 1 month)** | | | | | | | | |
| | 0,995 | 0,989 | 1,002 | 0,140 | 0,997 | 0,990 | 1,004 | 0,411 |

ND: NO Answer/Don't Know/not defined. OR: Odds Ratio. ALT: Alanine aminotransferase.

that underscores the negative impact of chronic HBV infection on HIV immunological response [20].

Although consistent with the results of some recent national and regional studies carried out in Uganda [8, 25], this prevalence was lower than the 17.9% reported in the general population in the same area in 2010 [11], the 10.8% prevalence found among pregnant HIV positive mothers in the same hospital in 2014 [26], and the 21.3% prevalence reported earlier across the

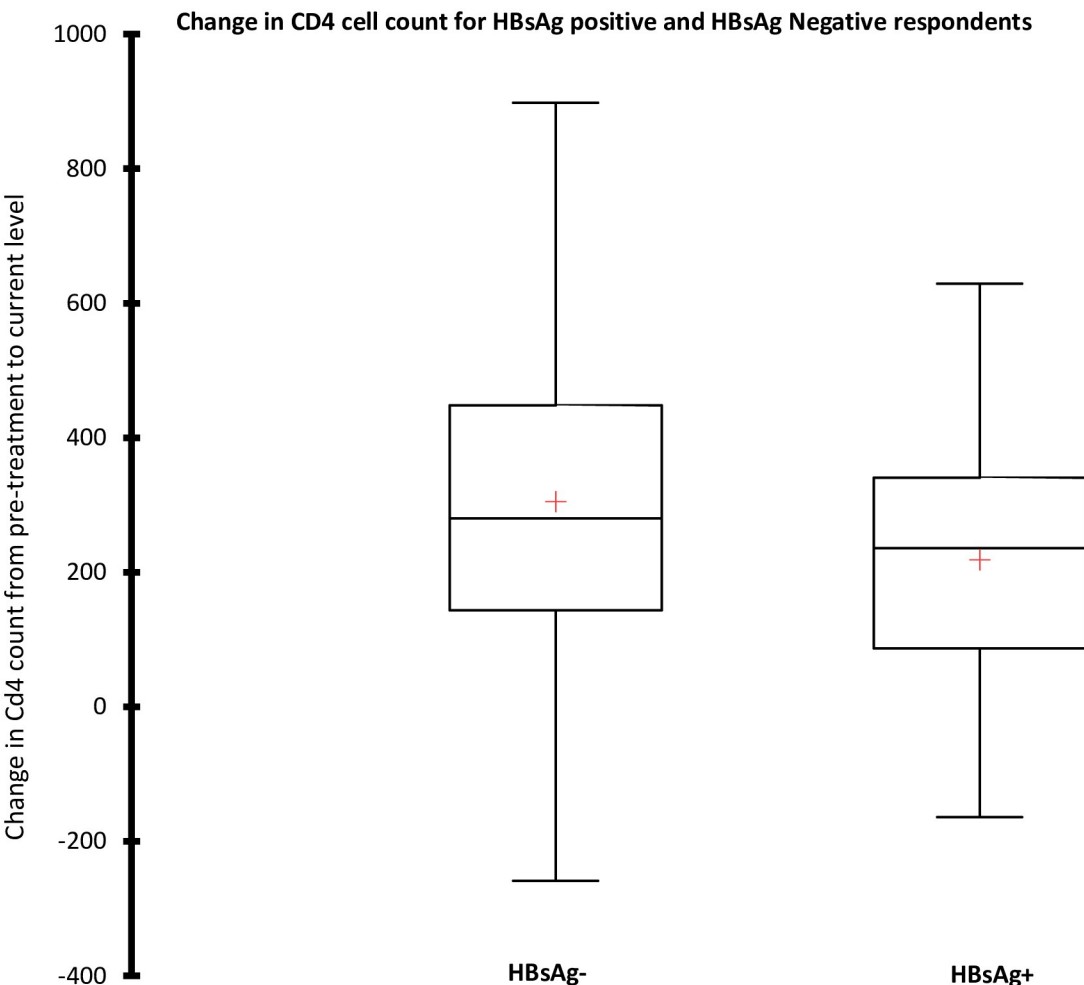

**Fig 1. Box plot showing the change in CD4+ T cell count from pre-ART initiation level to the most recent in HBsAg positive and negative patients.**

**Table 4. Predictors of CD4+ T cells response to ART therapy in the population of 950 PLWH (general linear model).**

|  | Beta | Standard error | Pr > \|t\| | Lower CI (95%) | Upper CI (95%) |
|---|---|---|---|---|---|
| CD4+ T cells-start of therapy | **-0,484** | 0,030 | **< 0,0001** | -0,543 | -0,425 |
| Months of antiretroviral therapy | **0,087** | 0,030 | **0,004** | 0,028 | 0,146 |
| Stayed in refugee camp (yes vs. no) | -0,017 | 0,030 | 0,575 | -0,075 | 0,041 |
| Gender (m vs. f) | **-0,154** | 0,030 | **< 0,0001** | -0,213 | -0,095 |
| HBV in the compound (yes vs. no) | 0,003 | 0,030 | 0,919 | -0,055 | 0,061 |
| HBV in the compound (ND vs. no) | -0,002 | 0,030 | 0,935 | -0,060 | 0,056 |
| Circumcision (yes vs. no) | -0,005 | 0,030 | 0,864 | -0,064 | 0,053 |
| HBsAg (yes vs. no) | **-0,065** | 0,029 | **0,027** | -0,123 | -0,008 |
| Tenofovir in start therapy (yes vs. no) | -0,047 | 0,030 | 0,114 | -0,106 | 0,011 |

ND: No answer/don't know/not defined.

whole Northern Uganda [6]. However, coinfection prevalence was similar to the prevalence of HBV co-infection in Europe [18, 27–29], North America [12, 30], and even in sub-Saharan Africa [31], where the reported ranges are wider than in Western countries and can reach 36% [5]. In countries where HBV is highly endemic, most cases of infection are attributable to vertical or horizontal transmission during early childhood [23, 32–34], although the findings of a recent study challenge the predominant role of vertical transmission in the area of Gulu [35]. This lower than expected prevalence raises the question as to whether there is a true decline in HBV prevalence in the region.

Our interviewers reported that many study participants had not heard of HBV until recently, thus most respondents did not know their maternal HBV infection status. However, it was found that being aware of the presence of HBV-infected subjects in the compound where a participant was brought up was independently associated with a higher risk of being HBsAg positive (P = 0.002).

Interestingly, having lived in an IDP camp was also an independent risk factor associated with HBV infection (P = 0.039). It is possible that life in these camps, which was characterised by greater crowding and higher risk sexual behaviours than regular village life, increased the likelihood of horizontal HBV transmission in our patients whose median age at the time ranged from 15 to 32 years. According to official data, more than 90% of Northern Uganda residents had lived in IDP camps by 2005 [10], but only 50% of the study participants reported it. This was possibly because those who had not been permanent residents (people who only spent nights in the camps and moved back to their own homes during the daytime) may have answered the question negatively.

The lower prevalence of HBV than that found in the general population of the same area deserves some comments. First of all, 77% of our study were women because 66% of the patients attending the AIDS clinic of St. Mary's Hospital are actually women [22]. This disproportion can be attributed to the practice of HIV testing during pregnancy [36] and the poor health-seeking behaviour of males [22]. In line with the findings of other African and European studies [5, 8, 27, 34, 37], the prevalence of HBV in our cohort was higher among men (11.7% *vs* 6.8%), but their under-representation may have lowered the actual overall prevalence. Secondly, our sampling procedures were performed in the AIDS clinic, and our study population consisted of regularly attending outpatients in good clinical condition with a high rate of adherence to ART (87.3%), which they had been receiving for a median of six years. Other studies of PLWH outpatients have reported comparable prevalence rates ranging from 6.7% to 11.5% [25, 26, 35] and, assuming that HIV/HBV co-infected patients experience a faster clinical evolution than other PLWH, it is possible that a disproportional number of co-infected patients were not enrolled [14, 27, 29]. Third, it cannot be excluded that mortality among the co-infected may have been higher than in the PLWH population as a whole, thus leading to a lower prevalence of HBV among the survivors. Nevertheless, the significant drop in prevalence noted in this and recent studies among different populations [8, 25] from values seen six to 12 years prior [6, 11] may signal the dawn of a decline in HBV prevalence in Northern Uganda.

The incidence of new chronic HBV infections is reduced in patients receiving lamivudine- and/or tenofovir-containing regimens and there is also a slim possibility of HBsAg seroreversion [38]. Despite this, the factors favouring horizontal transmission of HBV in our study population which is probably more relevant than vertical transmission suggest that consideration should be given to extending vaccine prophylaxis to HIV patients at risk.

The sensitivity and specificity of HBsAg rapid tests have recently been evaluated in a systematic review [31], which concluded that some perform better in the presence of high HBV viremia levels. As a potential limitation, the test we used was not included in this review and so

we do not know whether it shares the same limitation. However, as the vast majority of our HBsAg-positive patients had long been treated with tenofovir, it is likely that most of our study population had low viremia levels. If the sensitivity of our test depends on HBV viremia levels which we did not perform, this may be a limitation of the study. Of note also, the Ugandan comparator studies used similar HBsAg tests.

Similar to other studies carried out in high [18, 20]- and low-income countries [39], we found that HBV-HIV co-infection was associated with a worse immunological response to ART. Moreover, shorter-lasting ART, a lower baseline CD4+ T cell count and male sex also had independently negative effects on immunological outcomes. Assuming poor health-seeking behaviour among men and a worse natural history of co-infection, all these factors seem to be strictly related to each other.

Finally, although they were not involved in HBV transmission in the study population, polygamy and blood transfusion were significantly frequent: the former because it is customary for Acholi men to have more than one official wife, the latter because *P. falciparum* malaria is widespread in the area with potential for consequent severe anaemia [22].

Our finding that having lived in an IDP camp was associated with higher HBV prevalence, (something that is conceivably common to all IDP camps in areas that are highly endemic for HBV), as well as the worse CD4+ T cell response to ART in HBV-HIV coinfected persons supports the need to extend preventive interventions, particularly HBV screening and vaccination, to people at risk in this kind of emergency setting or in high HBV prevalence areas, coupled with prompt initiation of appropriate treatment for those positive. We also recommend further research to confirm the apparent drop of HBV prevalence in the Northern region of Uganda.

## Supporting information

**S1 Dataset. Data_HBV_HIV_Uganda.**
(XLSX)

**S1 File. Questionnaire.**
(PDF)

**S2 File. Translated questionnaire.**
(PDF)

## Acknowledgments

We would like to thank Awacango Grace, Apaco Rose Okumu, Acayo Mercy Otim, Okello Denis, Apio Lillian Grace, and all of the people working at the AIDS clinic for their help in data collection. We would also like to thank Mary Ann Gleason and Tuppin Scrase for their support during the data collection procedures. We are grateful to Nicolas Laing who proof-read this work.

## Author Contributions

**Conceptualization:** Annacarla Chiesa, Emmanuel Ochola, Giuliano Rizzardini, Massimo Galli.

**Data curation:** Annacarla Chiesa, Emmanuel Ochola.

**Formal analysis:** Annacarla Chiesa, Emmanuel Ochola, Letizia Oreni.

**Funding acquisition:** Annacarla Chiesa, Giuliano Rizzardini.

**Investigation:** Annacarla Chiesa, Emmanuel Ochola.

**Methodology:** Annacarla Chiesa, Emmanuel Ochola, Giuliano Rizzardini, Massimo Galli.

**Project administration:** Annacarla Chiesa, Emmanuel Ochola.

**Resources:** Annacarla Chiesa, Emmanuel Ochola, Giuliano Rizzardini.

**Software:** Letizia Oreni, Paolo Vassalini.

**Supervision:** Annacarla Chiesa, Emmanuel Ochola, Giuliano Rizzardini, Massimo Galli.

**Validation:** Annacarla Chiesa, Emmanuel Ochola, Letizia Oreni, Giuliano Rizzardini, Massimo Galli.

**Visualization:** Annacarla Chiesa, Emmanuel Ochola, Paolo Vassalini.

**Writing – original draft:** Annacarla Chiesa, Emmanuel Ochola, Letizia Oreni, Paolo Vassalini, Giuliano Rizzardini, Massimo Galli.

**Writing – review & editing:** Annacarla Chiesa, Emmanuel Ochola, Letizia Oreni, Paolo Vassalini, Giuliano Rizzardini, Massimo Galli.

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
