## [Decision Letter · Decision Letter 0]

6 Aug 2020

PONE-D-20-20265

Hepatitis B and HIV coinfection in Northern Uganda: is a decline in HBV prevalence on the horizon?

PLOS ONE

Dear Dr. Ochola,

Thank you for submitting your manuscript to PLOS ONE. After careful consideration, we feel that it has merit but does not fully meet PLOS ONE’s publication criteria as it currently stands. Therefore, we invite you to submit a revised version of the manuscript that addresses the points raised during the review process.

We look forward to receiving your revised manuscript.

Kind regards,

Jason Blackard, PhD

Academic Editor

PLOS ONE

Journal Requirements:

2. Thank you for stating in the text of your manuscript "Written informed consent was obtained from all the participants or, in the case of those aged 13-17 years, their legal guardians". Please also add this information to your ethics statement in the online submission form.

3. We note that part of your methods state that patients were selected starting on 23 May 2016, whereas earlier in the methods April - June 2016 is cited. Please clarify the date your study began.

4. Please provide more details on the limitations of your study methods.

5. We note that you state "CD4+ cell counts were significantly lower in the HBV patients (p=0.025)" however Table 2 shows the opposite. Please revise your table if, for example, values were swapped accidentally.

6. Please include additional information regarding the survey or questionnaire used in the study and ensure that you have provided sufficient details that others could replicate the analyses. For instance, if you developed a questionnaire as part of this study and it is not under a copyright more restrictive than CC-BY, please include a copy, in both the original language and English, as Supporting Information.

7. In your Data Availability statement, you have not specified where the minimal data set underlying the results described in your manuscript can be found. PLOS defines a study's minimal data set as the underlying data used to reach the conclusions drawn in the manuscript and any additional data required to replicate the reported study findings in their entirety. All PLOS journals require that the minimal data set be made fully available. For more information about our data policy, please see http://journals.plos.org/plosone/s/data-availability.

Additional Editor Comments (if provided):

This is a cross-sectional study of HBV infection among persons living with HIV in Uganda.  Given the burden of viral hepatitis in sub-Saharan Africa, the need for such studies is high.

The population size is good at >950 individuals.  The overall prevalence of 7.9% is what would be expected for a country in sub-Saharan Africa.  However, this manuscript would benefit from careful review by a native English speaker and/or a professional editing service.

The authors should clarify what population reference 10 was conducted in . . . was this the general population or persons living with HIV?

Was the data collection questionnaire self-administered or conducted by a researcher or clinician?

How was HIV treatment adherence reported and confirmed?

The authors should comment on how many individuals receiving ART were receiving HBV-active drugs as part of their ART regimen.  These data on specific ART regimens is confusing.  Tenofovir is mentioned but what about 3TC?  Are they always given together or could some individuals receive tenofovir only or 3TC only?

It appears that HBV DNA testing was not conducted.  This and the lack of information on HBV genotypes should be mentioned explicitly as limitations in the discussion.

Reviewers' comments:

Reviewer's Responses to Questions

**Comments to the Author**

1. Is the manuscript technically sound, and do the data support the conclusions?

Reviewer #1: Partly

Reviewer #2: Yes

2. Has the statistical analysis been performed appropriately and rigorously? 

Reviewer #1: I Don't Know

Reviewer #2: Yes

3. Have the authors made all data underlying the findings in their manuscript fully available?

Reviewer #1: Yes

Reviewer #2: Yes

4. Is the manuscript presented in an intelligible fashion and written in standard English?

Reviewer #1: Yes

Reviewer #2: Yes

5. Review Comments to the Author

Reviewer #1: The study determined the prevalence, correlates of the risk of HBV infection, and the effect of co-infection on the outcomes of ART among people living with HIV in Northern Uganda. The study enrolled 1000 participants and screened for HBsAg using rapid ELISA. An HIV/HBV coinfection prevalence of 7.9% was reported in participants who were on ART. Majority of the participants were on a Tenofovir containing. The study filled an important gap in the field. I would like to commend the authors. The study was well conducted and the article well written. There are some minor changes and suggestions which might improve the study.

Major revision

1. The study speaks of a decline in HBV prevalence based on comparison with a previous population based study. However, there were differences in the studied populations. The population in the recent study was mostly on Tenofovir a potent anti-HBV drug for a median duration of 6 years. It is expected that the prevalence will be lower due to treatments effects as some patients might have lost the HBsAg. The population based 2010 survey might have included HIV negative participants who were not on HBV active treatment or people living with HIV but not on HAART. The ‘decline would have been better ascertained if it was compared to previous treatment experienced patients in the same population. Furthermore, the study admits to the possibility of the HBsAg underestimation due to the kit used, which was not reviewed for performances in low HBV levels. The prevalence reported was also similar to studies in similar populations elsewhere.

Minor revisions

Abstract

2. People living with HIV/AIDS (PLWHA) should be changed to ‘People living with HIV (PLWH)’ according to the NIAID HIV Language Guide (February 2020)

3.The study type and time of study enrolment is missing.

4. HbsAg under Patients and Method should be changed to HBsAg

5. Shored in results section should be corrected to shared

6. CD4+ cell should be corrected to CD4+ T cell here and elsewhere in the article.

Introduction

7. WHO should be written in full since it’s first mention.

8. There is a space missing between the sentence (page 3) ‘Furthermore, as Northern Uganda was the theatre of a civil war from 1989 to 2006’ and the reference.

9. HIV clients should be changed to people living with HIV (PLWH)

Materials and methods

10.Page 5, In the sentence A questionnaire was administered to collected information’, the word collected should be corrected to collect

11. Page 5, AST and ALT should be written in full at first mention.

12. Page 6: The authors mention that testing was done ‘in accordance with the standard procedures of St. Mary’s Hospital Lacor’. Are the procedures different from the manufacture’s protocols? If they are then the differences should be noted for ease of reproducibility since the hospital’s procedures were not referenced.

13. In the sentence ‘We also made CD4+ T cell counts’, made should be changed to measured

Results

14. The authors mention that the 35 subjects excluded were not going to affect results but did not qualify the statement as to why/how they were not going to affect results.

15. Include explanations of all abbreviations below tables.

16. Page 10. The authors states that ‘All of the variables with a P-value of >0.2 at univariate analysis were included in a multivariate analysis, but table 3 includes variables with p values which were at univariate analysis < 0.2

17. Table 4 is not uniform. Some variable are written in all capital letters while others are written in sentence case.

Discussion

The discussion is well written with sound conclusions.

Reviewer #2: This manuscript reports on the findings of a study aimed at exploring a potential shift in the prevalence of chronic hepatitis B virus (HBV) infection (based on the prevalence of HBsAg) among people living with HIV and AIDS (PLWHA) in Northern Uganda. Given intensified global efforts towards elimination of viral hepatitis by 2030, the findings of this study could help inform public health strategies particularly in endemic regions such as sub-Saharan Africa. I have the following comments on the manuscript that need addressing;

Major revisions

In the Discussion section on page 14, the authors state the following;

“This leaves a new question to confirm: is there a true decline in HBV prevalence in the region?”.

To better address the hypothesis of the study, could the authors clearly comment on how the following could have impacted on the prevalence of HBsAg found within the study population:

•The study made use of a health facility-based population that may have different health-seeking behaviour from PLWHA within the general population. In addition, the fact that 86.8% of the study population were on long-term HBV active regimens (3TC and TDF) could reduce the prevalence of HBsAg.

•It is well established that the burden of occult HBV infection (OBI) is higher among PLWHA than the general population. Given that the prevalence of OBI was not assessed as part of this study, could the burden of HBV infection have been underestimated – could the lack of testing for anti-HBc and HBV DNA been a limitation to fully understanding the proportion of PLWHA who had not had been infected with HBV?

Minor revisions

•Abstract

o“…selected to undergo a rapid hepatitis B surface antigen (HbsAg) after administering a questionnaire.” Insert the word “test” after “HBsAg” for better clarity.

o“…and having shored housing with HBV-infected people…” Do the authors mean “shared”?

•Introduction

oPage 3; "About 5-15% of persons living with HIV worldwide have hepatitis B virus (HBV) infection". The current Global Hepatitis Report (2017) compiled by the WHO estimates that the global prevalence of HBV infection in HIV-infected persons is 7.4%. I would suggest this as a more appropriate and up-to-date reference.

oPage 3; “…and a high prevalence in Chad, Cameron, and western Africa (≥8%)…” Do the authors mean “Cameroon”?

oPage 4; “…co-infected subjects more rapidly develop liver fibrosis…and respond less to HBV vaccine.” Could the authors elaborate on this for better clarity? Given that the hepatitis B vaccine is a preventative and not a therapeutic vaccine, it would not be administered to those who are already infected.

•Results

oPage 9; “…almost all the recruited participants were receiving ART (950/985)…” In the methods sections, the authors clearly indicate that all 35 participants who were yet to initiate ART had been excluded from analysis.

oIn Table 2, does the line item “Months of therapy” refer to the number of months participants have been on ART or some other form of therapy?

oPage 10; “…and this was independently associated a higher CD4+ T cell count at the time…” insert the word “with” after “associated”.

•Discussion

oPage 15; “…crowding and promiscuity than that associated with village life…” I would suggest that the authors replace the word "promiscuity" with "high risk sexual behaviour" if this is indeed what they are referring to.

oPage 16; “Finally, although they were not involved in HBV transmission…customary for Acholi men to have more than one official wife, the latter because P. falciparum malaria is widespread in the area.” If these findings have no bearing on the burden of HBsAg or the risk of HBV transmission within the population, I would suggest that the authors provide some clarity as to why it has been highlighted in the discussion section.

6. PLOS authors have the option to publish the peer review history of their article (what does this mean?). If published, this will include your full peer review and any attached files.

Reviewer #1: No

Reviewer #2: **Yes: **Edina Amponsah-Dacosta

---

## [Author Response · Author response to Decision Letter 0]

30 Sep 2020

We thank the reviewers for the very constructive comments to improve our paper.

We have attached all the responses in the "Response to reviewers" file. Each issue is specifically responded to, and reference is made to enable such changes to be located on the tracked and unmarked manuscripts. Here below we only list the issues and responses, but the table referred above gives a more organised response.

Thank you again

Dr. Emmanuel Ochola (on behalf of the authors)

EDITOR'S COMMENTS

1. Please ensure that your manuscript meets PLOS ONE's style requirements, including those for file naming

Response (RS):

The file and supporting information naming as well as style have been modified to fit Plos One requirements.

2. Thank you for stating in the text of your manuscript "Written informed consent was obtained from all the participants or, in the case of those aged 13-17 years, their legal guardians". Please also add this information to your ethics statement in the online submission form

RS

This information has been added to the ethics statement in the online submission form.

3. We note that part of your methods state that patients were selected starting on 23 May 2016, whereas earlier in the methods April - June 2016 is cited. Please clarify the date your study began

RS

We apologize for this mistake. The actual dates of interview and HBsAg sampling were from 12th April to June 17th. The mistake in dates have now been corrected.

4. Please provide more details on the limitations of your study methods

RS

The limitations of the HBsAg underestimating the prevalence is not unique to this, but also applies to similar prevalence studies in Uganda. The HIV specific population and treatment on tenofovir potentially lowering prevalence have been discussed in a paragraph under discussion

5. We note that you state "CD4+ cell counts were significantly lower in the HBV patients (p=0.025)" however Table 2 shows the opposite. Please revise your table if, for example, values were swapped accidentally

RS

Thanks for noting this. However, the only time that CD4 was higher for the HBV positive than the HBsAg negative was at baseline. In all the other times after ART initiation, the CD4_+ cell count was higher for the HBsAg positive. The p value of 0.025 was noting that the rise in CD4+ count from baseline to current level is higher for HBsAg negative participants. No change has been made

6. Please include additional information regarding the survey or questionnaire used in the study and ensure that you have provided sufficient details that others could replicate the analyses. For instance, if you developed a questionnaire as part of this study and it is not under a copyright more restrictive than CC-BY, please include a copy, in both the original language and English, as Supporting Information

RS

The two versions of the questionnaire have been added as supporting information

7. In your Data Availability statement, you have not specified where the minimal data set underlying the results described in your manuscript can be found. PLOS defines a study's minimal data set as the underlying data used to reach the conclusions drawn in the manuscript and any additional data required to replicate the reported study findings in their entirety. All PLOS journals require that the

minimal data set be made fully available. For more information about our data policy, please see

http://journals.plos.org/plosone/s/data-availability

RS

We thank the Editor for these comments.

We had earlier submitted the data as additional supporting information, in line with Plos recommendations. In this resubmission, we attach the final combined dataset as supporting information. However, we have not deposited it in any other specific online location.

8. Upon re-submitting your revised manuscript, please upload your study’s minimal underlying data set as either Supporting Information files or to a stable, public repository and include the relevant URLs,

DOIs, or accession numbers within your revised cover letter. For a list of acceptable repositories,

please see http://journals.plos.org/plosone/s/data-availability#loc-recommended-repositories. Any potentially identifying patient information must be fully anonymized

RS

9. Specific identifying materials have been removed and dataset attached

The population size is good at >950 individuals. The overall prevalence of 7.9% is what would be expected for a country in sub-Saharan Africa. However, this manuscript would benefit from careful review by a native English speaker and/or a professional editing service

RS

We thank you for the kind comment about sample size. Our point in this paper is that for the specific higher burden Northern Ugandan region. The described prevalence is lower than previous findings.

Proof reading (not professional editing) has been done leading to changes in sentence constructions, flow and edits in the document. We have duly acknowledged this service

10. The authors should clarify what population reference 10 was conducted in . . . was this the general population or persons living with HIV?

RS

The referenced study was done in the general population, with and without HIV. A statement “with no documentation of HIV status” has been added

11.Was the data collection questionnaire self-administered or conducted by a researcher or clinician?

RS: The questionnaires were administered by nurses trained as research assistants. All of them had experience in HIV care. A statement to that effect has been added

12. How was HIV treatment adherence reported and confirmed?

RS: Adherence was measured using a composite of self-reporting and pill count, or recorded as assessed and documented by the primary health provider

13. The authors should comment on how many individuals receiving ART were receiving HBV-active drugs as part of their ART regimen. These data on specific ART regimens is confusing. Tenofovir is mentioned but what about 3TC? Are they always given together or could some individuals receive tenofovir only or 3TC only?

RS: The Primary backbone regimen in use in Uganda has been Tenofovir/Lamivudine fixed dose combination, usually in addition to EFV or NVP. All the patients an ART the clinic actually take Lamivudine in combination with other drugs. We did not consider Lamivudine combined with Abacavir or Zidovudine to be an optimal treatment- as guided by the Uganda HIV treatment guidelines. 86.84% percent were on Lamivudine/Tenofovir Diopropyl Fumerate and 13.16% were on Lamivudine combined with zidovudine or Abacavir, as part of their Antiretroviral treatment.

Finally, we did not mention treatment with lamivudine-only because of the low rate of HBsAg seroconversion seen with this regimen, which may not change the results of our study

14. It appears that HBV DNA testing was not conducted. This and the lack of information on HBV genotypes should be mentioned explicitly as limitations in the discussion

RS: Given the prohibitive cost of HBV DNA testing and the high number of positive respondents, we were unable to provide testing. We have added a statement about the potential better performance of HBsAg in high HBV viraemia, as a limitation under Discussion section

REVIEWER 1

The study speaks of a decline in HBV prevalence based on comparison with a previous population based study. However, there were differences in the studied populations. The population in the recent study was mostly on Tenofovir a potent anti-HBV drug for a median duration of 6 years. It is expected hat the prevalence will be lower due to treatments effects as some patients might have lost the

HBsAg. The population based 2010 survey might have included HIV negative participants who were not on HBV active treatment or people living with HIV but not on HAART. The ‘decline would have been better ascertained if it was compared to previous treatment experienced patients in the same population. Furthermore, the study admits to the possibility of the HBsAg underestimation due to the

kit used, which was not reviewed for performances in low HBV levels. The prevalence reported was also similar to studies in similar populations elsewhere 

RESPONSE (RS): The assertions about a decline in prevalence compares many previous studies to current ones, not just the 2010 study, as articulated under discussion, lines 225 to 229. 

As reported in the “Discussion” paragraph, we are aware that it is not possible to compare a prevalence study conducted on general population with one conducted on an HIV-positive group. However, data about Northern Uganda are still few, so we had to use all the available sources.

We discussed why our population may have biases with the general population that explains this result (gender disproportion, we excluded all patients with critical conditions, we recruited potentially more adherent). 

Although it is true that long term TDF therapy may lead to HBsAg seroconversion, as stated in 2017 EASL guidelines this is a really rare event (3% after 48-52 weeks of TDF treatment in HBeAg positive patients and 0% after 48-52 weeks of TDF treatment in HBeAg negative patients). The bias is possible but of minor entity. Thus the decline may not be negligible

2. People living with HIV/AIDS (PLWHA) should be changed to ‘People living with HIV (PLWH)

RS: 2. Thanks for the suggestion. This has been corrected in 10 instances and now appears as PLWH

3. Study type and time of enrolment is missing

RS: Study type: cross sectional study, and period (April to June) now included. This has been corrected, page 2 lines 30 and 33

4. HbsAg under Patients and Method should be changed to HBsAg

RS: Thanks for this. Four instances were corrected in text, and multiple instances in the tables.

5. Shored in results section should be corrected to shared

RS: Shored now appears as shared

6. CD4+ cell should be corrected to CD4+ T cell here and elsewhere in the article. 

RS: Thanks for the suggestion. We have corrected multiple incidences in the text and tables

Introduction

7. WHO should be written in full since it’s first mention.

RS: Thanks, this has been corrected

8. There is a space missing between the sentence (page 3) Uganda was thetheatre of a civil war from 1989 to 2006’ and the reference.

RS: This space has now been included

9. HIV clients should be changed to people living with HIV (PLWH)

RS: Thanks for these, which have been corrected as shown under 2 above

Materials and methods

10. Page 5, In the sentence A questionnaire was administered to collected information’, the word collected should be corrected to collect

RS: We appreciate this comment. The word “collected” now appears a s “collect”

11. Page 5, AST and ALT should be written in full at first mention.

RS: Corrected as suggested

12. Page 6: The authors mention that testing was done ‘in accordance with the standard procedures of St. Mary’s Hospital Lacor’. Are the procedures different from the manufacture’s protocols? If they are then the differences should be noted for ease of reproducibility since the hospital’s procedures were not referenced.

RS: The Standard operating procedures do not vary. We have now added “in accordance to the standard operating procedures of the manufacturers adapted at St. Mary’s hospital Lacor

13. In the sentence ‘We also made CD4+ T cell counts’, made should be changed to measured 

RS The word measured is now in place of made

Results

14. The authors mention that the 35 subjects excluded were not going to affect results but did not qualify the statement as to why/how they were not going to affect results

RS: Because of the sample size, the results of a statistical analysis conducted on 985 or 950 subjects did not gave significantly different results. Consequently, we decide to present in the paper the analysis on the 950 patients undergoing ART to have a more homogeneous sample to discuss

15. Include explanations of all abbreviations below tables.

RS: These have now been included where elsewhere missing.

16. Page 10. The authors states that ‘All of the variables with a P-value of >0.2 at univariate analysis were included in a multivariate analysis, but table 3 includes variables with p values which were at univariate analysis < 0.2

RS: Thanks for noting this mistake, which has now been corrected to p values <0.2%

17. Table 4 is not uniform. Some variable are written in all capital letters while others are written in sentence case.

RS: Table 4 now corrected to have uniform non capitalized letters

Discussion

The discussion is well written with sound conclusions.

RS: Thanks for the encouraging comment

REVIEWER 2

MAJOR REVISIONS

In the Discussion section on page 14, the authors state the following;

“This leaves a new question to confirm: is there a true decline in HBV prevalence in the region?”.

To better address the hypothesis of the study, could the authors clearly comment on how the following could have impacted on the prevalence of HBsAg found within the study population:

•The study made use of a health facility-based population that may have different health-seeking behaviour from PLWHA within the general population. In addition, the fact that 86.8% of the study population were on long-term HBV active regimens (3TC and TDF) could reduce the prevalence of HBsAg.

•It is well established that the burden of occult HBV infection (OBI) is higher among PLWHA than the general population. Given that the prevalence of OBI was not assessed as part of this study, could the

burden of HBV infection have been underestimated – could the lack of testing for anti-HBc and HBV DNA been a limitation to fully understanding the proportion of PLWHA who had not had been

infected with HBV?

RS: 

Similar concerns have been answered under response to major revisions by Reviewer 1.

Most of the discussion focuses on the prevalence found.

The seeming decline in prevalence seen in this study is not an isolated finding. First, the population prevalence of HIV at the time was about 8.3%, and this is from the whole region.

Because of the low rate of HBsAg seroconversions anticipated with 3TC/TDF, our result should not be heavily biased by those therapy. 

Of course the lack of data about HBV-DNA and HBcAb is a limitation, but

even when we anticipate OBI to reduce prevalence in HIV, the prevalence is actually lower than previous (Bwogi et al, Ochola et al), but higher than a similar study done around the same time in the same region (UPHIA report).A previous study conducted among women attending antenatal care (ANC) of the same area found HBV positive among 10.81% HIV positive pregnant women. Such leaves the question on decline as something to be confirmed.

Lack of HBV DNA testing has been noted as a limitation answered above under the last comment by the Editor, as well as to Reviewer 1.

MINOR REVISIONS

Abstract

1. “…selected to undergo a rapid hepatitis B surface antigen (HbsAg) after administering a questionnaire.” Insert the word “test” after “HBsAg” for better clarity.

RS: Thanks for the suggestion, “Test” has been inserted

2.“…and having shored housing with HBV-infected people…” Do the authors mean “shared”?

RS: The right word “shared” now appears in place of shored

Introduction

3.Page 3; "About 5-15% of persons living with HIV worldwide have hepatitis B virus (HBV) infection".

The current Global Hepatitis Report (2017) compiled by the WHO estimates that the global prevalence of HBV infection in HIV-infected persons is 7.4%. I would suggest this as a more appropriate and upto-date reference.

RS: We thank the reviewer for this suggestion, which has now been integrated in our references

4.Page 3; “…and a high prevalence in Chad, Cameron, and western Africa (≥8%)…” Do the authors mean “Cameroon”?

RS: Thanks. We meant Cameroon, now correctly written

5. Page 4; “…co-infected subjects more rapidly develop liver fibrosis…and respond less to HBV vaccine.” Could the authors elaborate on this for better clarity? Given that the hepatitis B vaccine is a preventative and not a therapeutic vaccine, it would not be administered to those who are already infected.

RS: What we describe is how PLWH respond to HBV infection or HBV vaccine. We agree that there would be no point in giving HBV vaccine to people already infected by HBV virus. In order to eliminate this misunderstanding, we modified the phrase to exclude vaccination

Results

6.Page 9; “…almost all the recruited participants were receiving ART (950/985)…” In the methods sections, the authors clearly indicate that all 35 participants who were yet to initiate ART had been excluded from analysis.

RS: The opening statement of results section does clarify on the exclusion of the 35 from analysis. They are not included in Table 2

7.In Table 2, does the line item “Months of therapy” refer to the number of months participants have been on ART or some other form of therapy?

RS: This has been corrected to read “months of antiretroviral therapy”, since it applies to duration on antiretroviral therapy

8..Page 10; “…and this was independently associated a higher CD4+ T cell count at the time…” insert the word “with” after “associated”.

RS: Thanks. “With” has now been included

Discussion

9.Page 15; “…crowding and promiscuity than that associated with village life…” I would suggest that the authors replace the word "promiscuity" with "high risk sexual behaviour" if this is indeed what they are referring to.

RS: Thanks for suggesting a more respectful word that has now replaced promiscuity

10. Page 16; “Finally, although they were not involved in HBV transmission…customary for Acholi men to have more than one official wife, the latter because P. falciparum malaria is widespread in the area.” If these findings have no bearing on the burden of HBsAg or the risk of HBV transmission within the population, I would suggest that the authors provide some clarity as to why it has been highlighted in the discussion section.

RS: Malaria is thought to predispose people to having severe anemia that might necessitate blood transfusion, a risk factor. These unstudied potential risk factors do exist in the community, hence the mention.

We have added that severe anemia is a potential consequence of P. falciparum infection. This is to support the reason for transfusion-based HBV risk

---

## [Decision Letter · Decision Letter 1]

19 Oct 2020

PONE-D-20-20265R1

Hepatitis B and HIV coinfection in northern Uganda: is a decline in HBV prevalence on the horizon?

PLOS ONE

Dear Dr. Ochola,

Thank you for submitting your manuscript to PLOS ONE. After careful consideration, we feel that it has merit but does not fully meet PLOS ONE’s publication criteria as it currently stands. Therefore, we invite you to submit a revised version of the manuscript that addresses the points raised during the review process.

Please make the minor updates requested by Reviewer 2 prior to acceptance of your manuscript.

We look forward to receiving your revised manuscript.

Kind regards,

Jason Blackard, PhD

Academic Editor

PLOS ONE

Additional Editor Comments (if provided):

Please make the minor updates requested by Reviewer 2 prior to acceptance of your manuscript.

Reviewers' comments:

Reviewer's Responses to Questions

**Comments to the Author**

1. If the authors have adequately addressed your comments raised in a previous round of review and you feel that this manuscript is now acceptable for publication, you may indicate that here to bypass the “Comments to the Author” section, enter your conflict of interest statement in the “Confidential to Editor” section, and submit your "Accept" recommendation.

Reviewer #1: All comments have been addressed

Reviewer #2: All comments have been addressed

2. Is the manuscript technically sound, and do the data support the conclusions?

Reviewer #1: Yes

Reviewer #2: Yes

3. Has the statistical analysis been performed appropriately and rigorously? 

Reviewer #1: Yes

Reviewer #2: Yes

4. Have the authors made all data underlying the findings in their manuscript fully available?

Reviewer #1: Yes

Reviewer #2: Yes

5. Is the manuscript presented in an intelligible fashion and written in standard English?

Reviewer #1: Yes

Reviewer #2: Yes

6. Review Comments to the Author

Reviewer #1: (No Response)

Reviewer #2: Line 32; “Factors associated with HBsAg positivity were analysed univariate...” Please insert the word, “using” before “univariate”.

Line 165; “viral load, below 20 copies/μ.…” Please confirm the unit of measure for the viral load assessment.

Lines 206-207; “…HIV infected patients in area busy hospital clinic in Northern Uganda...” This sentence is unclear to me, kindly rephrase for better clarity.

7. PLOS authors have the option to publish the peer review history of their article (what does this mean?). If published, this will include your full peer review and any attached files.

Reviewer #1: **Yes: **Motswedi Anderson

Reviewer #2: **Yes: **Edina Amponsah-Dacosta

---

## [Author Response · Author response to Decision Letter 1]

27 Oct 2020

Reviewer 2

1. Line 32; “Factors associated with HBsAg positivity were analysed univariate...” Please insert the word, “using” before “univariate”.

RESPONSE: We thank the reviewer for spotting this. We have now added the word “using” to make the statement understandable. This correction can be found on page 6 line 132 in both the tracked and clean documents.

2. Line 165; “viral load, below 20 copies/μ.…” Please confirm the unit of measure for the viral load assessment.

RESPONSE: The unit of viral load is copies per milliliters (ml). Therefore, ml has now replaced µ. Thank you very much. This is reflected in the tracked document on page 9 line 165 in the tracked document and on page 9 line 164 in the clean document.

3. Lines 206-207; “…HIV infected patients in area busy hospital clinic in Northern Uganda...” This sentence is unclear to me, kindly rephrase for better clarity

RESPONSE: Thank you for noting this. The word area was supposed to be only “a”. The phrase now reads, “This study found that HBV infection prevalence among HIV infected patients in a busy hospital clinic in Northern Uganda was 7.9%”. Page 14 line 206 in the tracked document and page 14 line 205 in the clean document.

---

## [Editor Report · Decision Letter 2]

30 Oct 2020

Hepatitis B and HIV coinfection in northern Uganda: is a decline in HBV prevalence on the horizon?

PONE-D-20-20265R2

Dear Dr. Ochola,

We’re pleased to inform you that your manuscript has been judged scientifically suitable for publication and will be formally accepted for publication once it meets all outstanding technical requirements.

Kind regards,

Jason Blackard, PhD

Academic Editor

PLOS ONE

Additional Editor Comments (optional):

None

Reviewers' comments:

None

---

## [Editor Report · Acceptance letter]

6 Nov 2020

PONE-D-20-20265R2 

Hepatitis B and HIV coinfection in northern Uganda: is a decline in HBV prevalence on the horizon? 

Dear Dr. Ochola:

I'm pleased to inform you that your manuscript has been deemed suitable for publication in PLOS ONE. Congratulations! Your manuscript is now with our production department. 

Kind regards, 

on behalf of

Dr. Jason Blackard 

Academic Editor

PLOS ONE